# Prevalence and factors associated with sarcopenia among older adults in a post-acute hospital in Singapore

**Charmaine Tan You Mei[1,2], Sharna Seah Si Ying[3], Doris Lim Yanshan[4], Siew Van Koh[4], Ganeshan Karthikeyan[4], Olivia Xia Jiawen[3], Xuan Lin Low[5], Hui Yi Quek[6], Andrea Ong Shuyi[1], Lian Leng Low[1,2], Junjie Aw[1,2]***

1 Post-Acute and Continuing Care Department, Outram Community Hospital, SingHealth Community Hospitals, Singapore, Singapore, 2 SingHealth Duke-NUS Family Medicine Academic Clinical Program, Singapore, Singapore, 3 Research and Translational Innovation Office, SingHealth Community Hospitals, Singapore, Singapore, 4 Rehabilitation Department, Outram Community Hospital, SingHealth Community Hospitals, Singapore, Singapore, 5 Department of Health and Social Science, Singapore Institute of Technology, Singapore, Singapore, 6 Department of Biological Sciences, National University of Singapore, Singapore, Singapore

* aw.junjie@singhealth.com.sg

**Data Availability Statement:** The data underlying the results presented in the study are available from SingHealth Community Hospitals HQ,

## Abstract

### Background

Sarcopenia is common in older adults worldwide, but its prevalence varies widely owing to differences in diagnostic criteria, population sampled, and care setting. We aimed to determine the prevalence and factors associated with sarcopenia in patients aged 65 and above admitted to a post-acute hospital in Singapore.

### Methods

This was a cross-sectional study of 400 patients recruited from a community hospital in Singapore. Data including socio-demographics, physical activity, nutritional status, cognition, clinical and functional status, as well as anthropometric measurements were collected. Sarcopenia was defined using the Asian Working Group for Sarcopenia 2019 criteria [AWGS2019].

### Results

Of the 383 patients with complete datasets, overall prevalence of sarcopenia was 54% while prevalence of severe sarcopenia was 38.9%. Participants with increased age, male gender and a low physical activity level were more likely to be sarcopenic, while those with higher hip circumference and higher BMI of ≥27.5m/kg2 were less likely to be sarcopenic. Other than the above-mentioned variables, cognitive impairment was also associated with severe sarcopenia.

### Conclusions

More than 1 in 2 older adults admitted to a post-acute hospital in Singapore are sarcopenic. There is an urgent need to address this important clinical syndrome burden and to

Research and Translational Innovation Office Senior Manager Mr. Kevin Chong. He can be contacted at kevin.chong@singhealthch.com.sg.

**Funding:** Charmaine Tan was the author who received the the 2021 SingHealth Regional Health System (RHS), Population-based, Unified, Learning System for Enhanced and Sustainable Health (PULSES) Grant ID: CGOCt21S06. Funders did not play any role in the study design, data collection and analysis, decision to publish or preparation of the manuscript. Funds were used to recruit interns for coordinating administrative duties of the study on a day to day basis with oversight from the P.I. and Co-Is. Part of the fund was used to purchase the BIA InBody S10 and also a laptop that is encrypted to perform data analysis together with acquisition of Stata software licence.

**Competing interests:** The authors have declared that no competing interests exist.

**Abbreviations:** 3-MinNS, 3-minute Nutritional Screening; 5CST, 5 times chair stand test; 6mGS, 6 meters Gait speed; ASM, Appendicular skeletal muscle mass; ASMI, Appendicular skeletal muscle mass index; AWGS, Asian working group for sarcopenia; BIA, Bioelectrical impedance analysis; CCI, Charlson comorbidity index; CI, Confidence interval; CMMSE, Chinese Mini Mental state examination; EWGSOP, European Working Group on Sarcopenia in Older People; GPAQ, Global Physical Activity Questionnaire; HC, Hip circumference; IDF, International Diabetes Federation; IQR, Interquartile range; MBI, Modified Barthel Index; OCH, Outram Community Hospital; PP, Physical performance; RDG, Rehabilitation diagnostic groups; SPPB, Short Physical Performance Battery; WC, Waist circumference; WHR, Waist hip ratio.

identify patients at risk of sarcopenia in post-acute settings in Singapore for early intervention.

## Introduction

The proportion of older adults worldwide is rapidly growing, with projected doubling by 2050 [1]. Close to seventeen percent of Singapore's residents were aged 65 and above in 2022, and this is rising 4–5.6% year on year [2]. Increased life expectancy globally is not equitable with healthy life expectancy [3]. Echoing this, the United Nations General Assembly has declared 2021–2030 the Decade of Healthy Ageing [4]. Increased age is associated with frailty and sarcopenia [5–8]. These geriatric syndromes herald increased healthcare utilization and costs from associated morbidity and mortality [9–12]. Among inpatients undergoing rehabilitation, sarcopenia had been reported to be associated with worse recovery of function and lower rate of home discharge in hospitalized adults undergoing rehabilitation [13].

Sarcopenia is common in older adults worldwide, but it's prevalence varies widely owing to differences in diagnostic criteria, population sampled, and care setting [14]. Sarcopenia prevalence reported in overseas studies was 26.9–58% in inpatient post-acute and rehabilitation wards and 50.9–60.2% in daycare facilities [11,15–18]. Sarcopenia has been reported to be associated with age, gender, marital status, comorbidities, smoking, physical activity, BMI, waist circumference, hip circumference, nutrition, and cognition [13,15–24].

Several studies in Singapore had reported sarcopenia prevalence among varying population groups in the community.

For example, one study reported sarcopenia prevalence of 76% among community ambulant adults aged 65 and above who were at medium or high risk of malnutrition [22]. The prevalence of sarcopenia in primary care and specialist outpatient clinics ranged from 27.4% in patients aged 60–89 years old with Type 2 Diabetes Mellitus using AWGS criteria, to 44% in patients aged 65 and above based on SARC-F questionnaire only [19,25]. Type 2 diabetes mellitus association with sarcopenia is also mirrored in an interesting outpatient study overseas where post-menopausal women with type 2 diabetes mellitus are 5 times more likely to have osteo-sarcopenia than those without type 2 diabetes mellitus [26].

However, till date there is none investigating sarcopenia prevalence using established recommended criteria in the inpatient post-acute setting. We aim to determine the prevalence of sarcopenia, severe sarcopenia and their associations in an inpatient post-acute hospital in Singapore.

## Materials and methods

### Setting

This cross-sectional study was conducted from May to November 2022 at Outram Community Hospital [OCH].

A community hospital in Singapore is a purpose built hospital to provide medical, nursing and rehabilitation care for patients who require a short period of continuation of care after their stay in the acute hospitals i.e. post-acute care before being discharged into the community [27].

Admissions into OCH come from the Singapore General Hospital [SGH], the largest co-located acute tertiary hospital in Singapore servicing the southeastern region of Singapore.

## Study participants and recruitment

All patients aged 65 years old and above admitted to OCH under subacute or rehabilitation service from 17 May to 16 November 2022 were consecutively screened based on eligibility criteria. Participants who were unable to understand English or Mandarin, refused to participate, or unable to give informed consent or follow instructions due to conditions such as neuropsychiatric or neurocognitive disease were excluded. Additional excluded participants included those who were terminally ill; medically unstable; had conditions which precluded sarcopenia assessment such as anyone with a cardiac pacemaker, an implantable defibrillator, amputation of limb[s], intravenous hydration, fluid overload and had temporary restrictions in weight-bearing status of upper or lower limbs.

Eligible participants were then recruited after written informed consent was obtained. All participants recruited were assigned to a unique serial number in order to delink and deidentify them. All data was collected during the inpatient stay in OCH, with no further follow-up or contact after discharge.

This study was approved by SingHealth Centralized Institutional Review board [CIRB Ref 2021/2817].

## Data collection

All interviewers received standardized training on interview techniques with trial interviews held with trainer to ensure fidelity. Face-to-face interviews with the patients were conducted in Mandarin and English.

Data collected included 1] demographics [age, sex, race, marital status, living setup, type of dwelling and highest educational qualification]; 2] physical activity and sedentary time; 3] nutritional status; 4] cognition; and 5] other clinical parameters [weight, height, smoking history, Modified Barthel Index [MBI] scores on admission, Charlson Comorbidity Index [CCI]; history of COVID-19 infection; waist circumference [WC]; hip circumference [HC] and Rehabilitation Diagnostic Groups [RDG] classification, an administrative framework to group patients admitted for rehabilitation so as to facilitate transition and collaboration across the stakeholders in the care spectrum.

Physical activity level and sedentary time were assessed using the Global Physical Activity Questionnaire [GPAQ], which was developed by WHO with standardized approach and question guide available both in English and Mandarin translations, and used in multiple Singapore studies including the National Health Survey 2010 and 2019 [21,28–33]. As per the GPAQ, participants' sedentary time in minutes per day and their physical activity level [low and moderate or high] were collected [34].

The 3-Minute Nutritional Screening [3-MinNS] tool, developed in Singapore and validated for medical and surgical inpatients in a Singapore hospital, was used to screen nutritional status [35,36]. Participants were grouped into low [0–2], moderate [3–4], or severe [5–9] malnutrition risk [35].

Cognitive assessment was done using the Chinese Mini Mental State Examination [CMMSE], which was also validated locally in English and Mandarin [37]. Those with CMMSE scores of 23 and below were considered as having impaired cognition [38].

The participants' inpatient medical records were accessed to retrieve information regarding the following: MBI on admission; CCI; history of COVID-19 infection; RDGs; weight [in kilogrammes] and height [in metres] on admission.

Body mass index [BMI] was calculated as body weight divided by square of height [kg/m$^2$]. Cut-offs were based on WHO BMI risk categories for cardiovascular disease and diabetes in Asian populations [39].

## Anthropometry

Waist circumference [WC] was measured using a stretch-resistant tape at the midpoint between the lower margin of the least palpable rib and the top of the iliac crest. Waist circumference cut-offs were determined as per The International Diabetes Federation [IDF] consensus worldwide definition for metabolic syndrome [40].

Hip circumference [HC] was measured around the widest portion of the buttocks [41]. Waist-hip ratio [WHR] was calculated by dividing WC over HC. Cut-offs for WHR were based on World Health Organization [WHO], redefining obesity—the Asia-Pacific perspective [42].

## Sarcopenia diagnostic criteria

There have been several proposed diagnostic algorithms for sarcopenia. In 2010, the European Working Group on Sarcopenia in Older People [EWGSOP] proposed the first practical clinical definition and diagnostic criteria for sarcopenia based on assessment of muscle mass, muscle strength and physical performance [43]. This was revised in 2018 to EWSOP2, which used low muscle strength as the primary parameter of sarcopenia [10]. To address differences in cut-off values of measurements in Asian populations from Europeans, an Asian consensus was derived by the Asian Working Group for Sarcopenia [AWGS] in 2014 and revised in 2019 [44,45].

In our study, sarcopenia was assessed and diagnosed using criteria as per AWGS2019 [45]. Sarcopenia was diagnosed in the presence of low muscle mass, with either low muscle strength or low physical performance [PP]. Those with low muscle mass, low muscle strength, and low PP, were further subclassified as "severe sarcopenia" for sub-group analyses [45].

## Muscle mass

**Muscle mass.** Appendicular skeletal muscle mass [ASM] was determined using a multifrequency bioelectrical impedance analysis [BIA] InBody S10 Body Composition Monitor. ASMI was calculated by dividing ASM by height squared [kg/m$^2$]. BIA measurements were conducted under standardised protocols i.e. before therapy sessions, in supine position with limbs abducted and ensuring no contact with metal frame of bed [46]. Low muscle mass was defined as appendicular skeletal muscle mass index [ASMI] <7kg/m2 in males and <5.7kg/m2 in females.

**Muscle strength.** Muscle strength was assessed via handgrip strength [HGS] which was measured using a dynamometer [BASELINE 12–0240 standard hydraulic hand dynamometer] in seated posture, with shoulder adducted and elbow flexed to 90 degrees and forearm in neutral as recommended by American Society of Hand Therapists [47]. The maximum reading from at least two trials using either hand in a maximum-effort isometric contraction was used for analysis [45]. Low muscle strength was defined as handgrip strength [HGS] <28kg in males and <18kg in females.

**Physical performance.** Low PP was defined as gait speed <1.0m/s over 6-metre walk [6mGS], or 5-time chair stand test [5CST] requiring ≥12s, or Short Physical Performance Battery [SPPB] score of ≤9.

6mGS was measured as time taken to walk 6m at a normal pace from a moving start, without deceleration, with average of two trials taken as the recorded speed [m/s] [45]. Walking aids were permitted if necessary and documented if it was used. 5CST was measured as the time taken to rise and sit five times as quickly as possible with no contact against the back of the chair and maximal extension of knees [48]. SPPB is a 3-part performance-based test of balance, gait speed, and 5CST, with a score of 0 to 12.

## Statistical analyses

Categorical variables were presented as proportions and continuous variables summarized as means+/- SD or medians with interquartile ranges [IQR [25th percentile, 75th percentile]] as appropriate. Pearson chi-square test was used to compare categorical variables and logistic regression to compare continuous variables as appropriate for univariate analyses.

Multivariate logistics regression was done with all variables included as co-variates to arrive at adjusted odds ratio with 95% confidence interval [CI] associated with sarcopenia outcomes. Similarly multinomial regression was done on all variables to calculate adjusted odds ratio associated with severe sarcopenia outcomes.

All analyses were two-tailed and p value <0.05 was considered statistically significant. Statistical analysis was performed using Stata Version 17 [StataCorp, College Station, TX, USA].

## Sample size estimation

Prevalence of sarcopenia in overseas studies on patients with a similar profile to our study population ranged between 26.9–60.2% [11,13,15,18]. In view of this large range in prevalence, it is recommended to take a conservative estimate of 50% for the expected prevalence, that will lead to the largest estimate for sample size [49].

Using the Finite Population Correction for prevalence for cross-sectional studies [as more than 5% of the population is being sampled, and the population has a known population size], sample size, n = $\frac{N[z^2]p[1-p]}{(d^2)(N-1)+(z^2)p[1-p]}$ [50].

z is the statistic corresponding to a confidence level of 95% and α of 0.05, which is 1.96; d is the level of precision which is selected to be 5% [as p is assumed to be between 10%-90%]; N is the finite population size, which is taken to be 1440 based on average of 240 admissions per month x 6 months of recruitment; p is the expected prevalence which is taken to be 50%, n is hence 304.

# Results

A total of 400 patients were recruited, of which 17 [4%] had incomplete data due to discharge or transfer to acute hospital for acute medical issues. Hence, 383 [96%] patients with complete data were analysed in this study. Fig 1 shows the process of identifying sarcopenia in our study population using AWGS2019 diagnostic algorithm.

## Prevalence of sarcopenia and severe sarcopenia

Prevalence of sarcopenia using AWGS 2019 criteria in our study population was 54.0% [n = 207] with 66.4% of males being sarcopenic and 45.9% of females being sarcopenic. Prevalence of severe sarcopenia in our cohort was 38.9% [n = 149].

## Characteristics of study population and unadjusted odds ratio

Table 1 illustrates the characteristics of our participants with corresponding unadjusted odds ratios of sarcopenia. The median age of participants was 75 years old [IQR: 70–81], with more females [60.3%] than male participants, and majority [93%] were Chinese. Slightly more than half [56.1%] were currently married. Majority [91.9%] live in public governmental housing [Housing Development Board [HDB] apartments]. About two thirds [60.4%] of our cohort have less than secondary school education and only 20% of participants were smokers or ex-smokers. 2/3 of our participants had CCI more than "0". 62% of participants were not known to be infected by COVID-19 prior to recruitment into study.

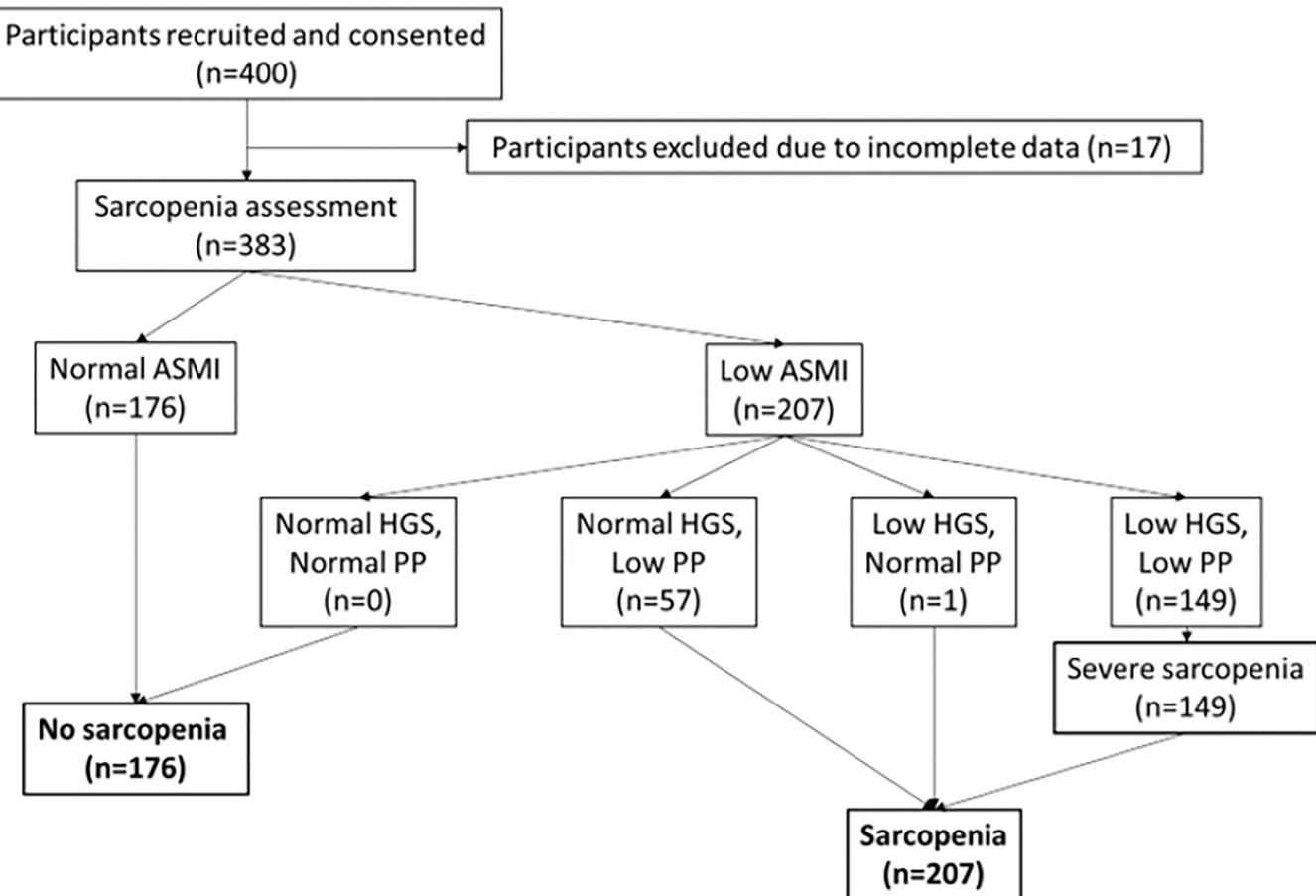

**Fig 1. Process of identifying sarcopenia in our study population using Asian Working Group for Sarcopenia (AWGS) 2019 criteria (ASMI: Appendicular skeletal muscle mass index; HGS: Handgrip strength; PP: Physical performance).**

Slightly more than half [56.4%] were at moderate or high risk of malnutrition. Physical activity level was evenly distributed with half [50.9%] of the participants assessed to have low physical activity level. Approximately half of our participants [49.9%] had CMMSE 23 or less. Slightly over half [57.4%] of participants had BMI of 23kg/m$^2$ or more, while approximately 2/3 had an elevated waist circumference and waist-hip ratio.

Half [54.1%] of the participants had an MBI score of 0–60 on admission, indicating severe to total dependency in activities of daily living. Participants were mostly admitted for rehabilitation [RDG] due to musculoskeletal conditions [48.8%], followed by deconditioning [18.5%], others [14.4%], hip fracture [11.5%], and stroke [5.1%].

## Factors associated with sarcopenia

Table 1 showed the univariable comparison between individuals with sarcopenia and without. Those with sarcopenia were older, more likely to be male, had higher CCI, had moderate or severe risk of malnutrition [compared to no risk] and low physical activity. They were also less likely to be currently married.

Table 2 showed the multivariable analysis. Older age [OR 1.06 [1.01–1.12]], male sex [OR 2.80 [1.12–7.02]] and low physical activity [OR 2.13 [1.17–3.89]] were associated with sarcopenia. Higher BMI more than or equal to 27.5 [OR 0.16 [0.05–0.52]] and greater hip circumference [OR 0.86 [0.81–0.92]] are inversely associated with sarcopenia [Table 2].

**Table 1. Characteristics of study population and factors associated with sarcopenia on univariate analysis.**

| Characteristic | Total [n = 384] | Non-Sarcopenic [n = 176] | Sarcopenia [n = 208] | Unadjusted OR [95% CI] | p-value |
|---|---|---|---|---|---|
| Age in years, median [IQR] | 75 [70, 81] | 74 [69, 78] | 77 [72, 82] | 1.09 [1.06–1.13] | <0.001 |
| Sex, n [%]<br>Male | 152 [39.6] | 51 [29.0] | 101 [48.8] | 2.34 [1.53–3.57] | <0.001 |
| Female | 231 [60.3] | 125 [71.0] | 106 [51.2] | 1 | |
| Race, n [%]<br>Chinese | 356 [93] | 160 [90.9] | 196 [94.7] | 1 | 0.291 |
| Malay | 16 [4.2] | 11 [6.25] | 5 [2.4] | 0.37 [0.13–1.09] | |
| Indian | 8 [2.1] | 4 [2.27] | 4 [1.9] | 0.82 [0.20–3.32] | |
| Others [2 Eurasians, 1 Sikh] | 3 [0.8] | 1 [0.6] | 2 [1.0] | 1.63 [0.15–18.17] | |
| Current Marital Status, n [%] | | | | | 0.039 |
| Single / Widowed / Divorced / Separated | 168 [43.9] | 67 [38.1] | 101 [48.8] | 1.55 [1.03–2.33] | |
| Currently married | 215 [56.1] | 109 [61.9] | 106 [51.2] | 1 | |
| Living Setup, n [%] | | | | | |
| Alone | 85 [22.2] | 33 [18.8] | 52 [25.1] | 1.45 [0.89–2.38] | 0.135 |
| With someone | 298 [77.8] | 143 [81.3] | 155 [74.9] | 1 | |
| Housing Type, n [%] | | | | | |
| HDB [public housing] | 352 [91.9] | 163 [92.6] | 189 [91.3] | 0.89 [0.42–1.88] | 0.622 |
| Condominium/Landed [private housing] | 30 [7.8] | 13 [7.4] | 17 [8.2] | 1 | |
| Others [overseas property] | 1 [0.3] | 0 [0] | 1 [0.5] | - | |
| Highest Educational Qualification, n [%]* | | | | | |
| Below GCE "N" or "O" levels | 230 [60.4] | 100 [56.8] | 130 [63.4] | 1.32 [0.87–1.99] | 0.189 |
| GCE "N" or "O" level and above | 151 [39.6] | 76 [43.2] | 75 [36.6] | 1 | |
| Smoking history, n [%] | | | | | |
| Never smoker | 308 [80.4] | 148 [84.1] | 160 [77.3] | 1 | 0.095 |
| Ex or Current smoker | 75 [19.6] | 28 [15.9] | 47 [22.7] | 1.55 [0.92–2.61] | |
| Charlson Comorbidity Index, n [%] | | | | | |
| 0 | 130 [33.9] | 72 [40.9] | 58 [28.0] | 1 | 0.022# |
| 1 and 2 | 173 [45.2] | 74 [42.1] | 99 [47.8] | 1.66 [1.05–2.63] | |
| 3 and more | 80 [20.9] | 30 [17.1] | 50 [24.2] | 2.07 [1.17–3.66] | |
| Ever infected by COVID-19 before recruitment into study, n [%] | | | | | |
| No | 237 [61.9] | 113 [64.2] | 124 [59.9] | 1 | 0.388 |
| Yes | 146 [38.1] | 63 [35.8] | 83 [40.1] | 1.20 [0.79–1.82] | |
| Malnutrition Risk n [%]* | | | | | |
| No risk | 170 [44.6] | 112 [64.0] | 58 [28.2] | 1 | <0.001 |
| Moderate risk | 124 [32.6] | 40 [22.9] | 84 [40.8] | 4.06 [2.48–6.63] | |
| Severe risk | 87 [22.8] | 23 [13.1] | 64 [31.1] | 5.37 [3.03–9.52] | |
| Physical Activity, n [%] | | | | | |
| Low | 195 [50.9] | 78 [44.3] | 117 [56.5] | 1.63 [1.09–2.45] | 0.017 |
| Moderate to high | 188 [49.1] | 98 [55.7] | 90 [43.5] | 1 | |
| Sedentary Time in minutes/day, median [interquartile range 25th centile, 75th centile] | 800 [690–895] | 800 [685–890] | 810 [710–900] | 1.00 [0.9997–1.0022] | 0.121 |
| Cognition, n [%] | | | | | |
| CMMSE Score 23 and less | 191 [49.9] | 73 [41.5] | 118 [57.0] | 1.87 [1.25–2.81] | 0.002 |
| CMMSE Score 24 and more | 192 [50.1] | 103 [58.5] | 89 [43.0] | 1 | |
| BMI[a], kg/m$^2$ [%] | | | | | |

*(Continued)*

**Table 1.** (Continued)

| Characteristic | Total [n = 384] | Non-Sarcopenic [n = 176] | Sarcopenia [n = 208] | Unadjusted OR [95% CI] | p-value |
|---|---|---|---|---|---|
| Less than 18.5 | 56 [14.6] | 3 [1.7] | 53 [25.6] | 4.31 [1.23–15.17] | <0.001 |
| 18.5–22.9 | 107 [27.9] | 21 [11.9] | 86 [41.6] | 1 | |
| 23–27.4 | 133 [34.7] | 74 [42.1] | 59 [28.5] | 0.19 [0.11–0.35] | |
| 27.5 and above | 87 [22.7] | 78 [44.3] | 9 [4.4] | 0.03 [0.01–0.07] | |
| Waist Circumference [b], cm [%] | | | | | |
| Males ≤90; Females ≤80 | 128 [33.4] | 20 [11.4] | 108 [51.2] | 8.51 [4.96–14.59] | <0.001 |
| Males >90; Females >80 | 255 [66.6] | 156 [88.6] | 99 [47.8] | 1 | |
| Hip Circumference in cm, mean [SD] | 94.2 [8.9] | 99.6 [7.9] | 89.6 [6.9] | 0.83 [0.79–0.86] | <0.001 |
| Waist-Hip Ratio [WHR][c], n [%] | | | | | |
| Males 1.0 and below Females 0.85 and below | 119 [31.1] | 35 [19.9] | 84 [40.6] | 2.75 [1.73–4.37] | <0.001 |
| Males more than 1.0 Females more than 0.85 | 264 [68.9] | 141 [80.1] | 123 [59.4] | 1 | |
| MBI score on admission, n [%] | | | | | |
| MBI 0–60 [severe to total dependency] | 207 [54.1] | 80 [45.5] | 127 [61.4] | 1.91 [1.27–2.86] | 0.002 |
| MBI 61–100 [moderate dependency to functional independence] | 176 [46.0] | 96 [54.6] | 80 [38.7] | 1 | |
| Admission RDG, n [%] | | | | | |
| Deconditioning | 71 [18.5] | 16 [9.1] | 55 [26.6] | 5.25 [2.80–9.85] | <0.001 |
| Others [e.g. IV antibiotics, wound care, malignancy] | 55 [14.4] | 16 [9.1] | 39 [18.8] | 3.72 [1.94–7.14] | |
| Stroke | 26 [6.8] | 9 [5.1] | 17 [8.2] | 2.88 [1.22–6.81] | |
| Hip fracture | 44 [11.5] | 22 [12.5] | 22 [10.6] | 1.53 [0.79–2.95] | |
| Musculoskeletal conditions | 187 [48.8] | 113 [64.2] | 74 [35.8] | 1 | |

[a] Cut-offs as per WHO BMI risk categories for cardiovascular disease and diabetes in Asian populations.

[b] Cut-offs as per IDF.

[c] Cut-offs as per WHO.

[#] p-value required for significance after Bonferroni correction is 0.0083.

[*]missing data n = 2.

### Factors associated with severe sarcopenia

Among the variables investigated, severe sarcopenia in our setting is associated with age [OR 1.10 [1.03–1.16]], males [OR 4.35 [1.5–12.64]], low physical activity [OR 2.78 [1.34–5.79]], and cognitive impairment with MMSE 23 or less [OR 2.26 [1.01–5.02]]. Higher BMI more than or equal to 27.5 [OR 0.20 [0.05–0.75]] and greater hip circumference [OR 0.84 [0.78–0.91]] are inversely associated with sarcopenia [Table 3].

## Discussion

We found a high prevalence of sarcopenia with more than 1 in 2 older adults admitted to an inpatient post-acute rehabilitation setting in Singapore having sarcopenia. In addition, what is worrisome is that 1 in 3 older adults inpatients have severe sarcopenia. Our sarcopenia prevalence is in keeping with worldwide literature where such post-acute units have the highest prevalence of sarcopenia followed by nursing home residents, then hospitalized older adults and lowest in community dwelling older adults [51]. This confirms the burden of disease in

**Table 2. Multivariable analysis on variables associated with sarcopenia.**

| Characteristic | Adjusted OR [95% CI] |
|---|---|
| Age, years | 1.06 [1.01–1.12] |
| Sex | |
| Male | 2.80 [1.12–7.02]* |
| Female | 1 |
| BMI, kg/m$^2$ | |
| <18.5 | 2.34 [0.56–9.67] |
| 18.5–22.9 | 1 |
| 23–27.4 | 0.53 [0.24–1.16] |
| 27.5 and above | 0.16 [0.05–0.52]*** |
| HC, cm | 0.86 [0.81–0.92]*** |
| Physical Activity level | |
| Low | 2.13 [1.17–3.89]* |
| Moderate to High | 1 |

P values

* <0.05

** <0.01

*** <0.001.

units performing rehabilitation and suggests that patients should be intervened earlier upstream even before they are being referred or admitted to such settings. There is also a certain urgency to identify this group of patients for further study and conduct interventions with the aim to reverse the sarcopenia.

Like many studies, we found that ageing is associated with sarcopenia [21–23,52]. This is in keeping with the observation where older adults lose muscle mass with increasing age with intramuscular and intermuscular fat infiltration [53–56]. Many underlying hypotheses for this had been proposed. Examples include role of chronic inflammation via biochemical pathways mediated by IL-1β, IL-6, TNF-α et cetera; hormonal changes with age such as reduced GH, IGF-1, testosterone and oestrogens; mitochondrial dysfunction; accessibility to nutrition and dietary patterns; lifestyle factors such as reduction in physical activity, obesity rates and smoking and last but not least, concurrent metabolic chronic diseases being more prevalent in older adults leading to the onset of sarcopenia [57–59]. However, interesting developments looking at birth weights and later onset sarcopenia seem to hint at other factors that go beyond ageing as contributory factors for sarcopenia, setting the stage for a comprehensive review at reversing sarcopenia at all levels [60,61].

Most global literature found a significant association between males and sarcopenia like our study although there are some studies showed an association between females and sarcopenia instead [18,19,21,22]. Nonetheless, this highlights likely unique gender differences in developments of sarcopenia [21,62]. It has been postulated that myostatin causing catabolism in males and reduced IGF-1 leading to anabolic decline in females have a role in the sex differences in sarcopenia development [63,64]. Other clinical parameters such as men losing muscle strength and testosterone faster than women also support the association of men being at higher odds of sarcopenia than women [65–69]. Future trials are needed to formulate sex specific interventions tailored to reverse sarcopenia so as to conclusively determine the causal effects of these associations.

Consistent with literature, low physical activity has an inverse association with muscle mass and higher odds of sarcopenia [70–72]. What remains debatable is designing an evidence-based exercise regime to reverse sarcopenia. The differential role of aerobic, resistance training and a combination of the types of physical activity may play different roles in preventing or

**Table 3. Multinomial regression on association factors with sarcopenia and severe sarcopenia.**

| Variables | Sarcopenia | | Severe sarcopenia | |
|---|---|---|---|---|
| | Adjusted OR [95% C.I.] | p-value | Adjusted OR [95% C.I.] | p-value |
| Age | 0.99 [0.93–1.06] | 0.829 | 1.10 [1.03–1.16] | 0.002 |
| Sex | | | | |
| Male | 1.57 [0.45–5.43] | 0.476 | 4.35 [1.50–12.64] | 0.007 |
| Female | Reference | | Reference | |
| Race | | | | |
| Chinese | - | | 0.15 [0.02–1.18] | 0.072 |
| Malay | - | | 0.28 [0.02–3.99] | 0.349 |
| Indian | Reference | | Reference | |
| Others [2 Eurasians, 1 Sikh] | - | | - | |
| Current Marital Status | | | | |
| Single/Widowed/Divorced/Separated | 1.46 [0.60–3.56] | 0.405 | 1.85 [0.83–4.13] | 0.136 |
| Married | Reference | | Reference | |
| Living Setup | | | | |
| Alone | - | | 3.23 [0.39–26.42] | 0.274 |
| With Tenant / Friend | Reference | | Reference | |
| With Family / Caregiver | - | | 2.37 [0.30–18.74] | 0.414 |
| Highest Educational Qualification* | | | | |
| Below GCE "N" or "O" levels | 1.81 [0.76–4.33] | 0.182 | 0.68 [0.30–1.51] | 0.34 |
| GCE "N" or "O" level and above | Reference | | Reference | |
| Smoking history | | | | |
| Never smoker | Reference | | Reference | |
| Ex or Current smoker | 1.09 [0.38–3.12] | 0.877 | 0.47 [0.18–1.20] | 0.113 |
| BMI | | | | |
| Less than 18.5 | 2.37 [0.42–13.34] | 0.329 | 3.65 [0.77–17.28] | 0.103 |
| 18.5–22.9 | Reference | | Reference | |
| 23–27.4 | 0.52 [0.20–1.41] | 0.2 | 0.53 [0.26–1.31] | 0.17 |
| 27.5 and above | 0.08 [0.01–0.50] | 0.007 | 0.20 [0.05–0.75] | 0.017 |
| Waist Circumference[a], cm | | | | |
| Males ≤90; Females ≤80 | 0.39 [0.11–1.37] | 0.142 | 0.76 [0.26–2.27] | 0.624 |
| Males >90; Females >80 | Reference | | Reference | |
| Hip Circumference in cm | 0.86 [0.78–0.94] | 0.001 | 0.84 [0.78–0.91] | <0.001 |
| Waist-Hip Ratio [WHR][b] | | | | |
| Males 1.0 and below Females 0.85 and below | 1.69 [0.45–6.31] | 0.432 | 1.28 [0.41–3.99] | 0.669 |
| Males more than 1.0 Females more than 0.85 | Reference | | Reference | |
| Physical Activity[c] | | | | |
| Low | 1.57 [0.71–3.48] | 0.263 | 2.78 [1.34–5.79] | 0.006 |
| Moderate to high | Reference | | Reference | |
| Sedentary Time in minutes/day | 1.00 [1.00–1.00] | 0.136 | 1.00 [1.00–1.00] | 0.474 |
| Malnutrition Risk | | | | |
| No risk | Reference | | Reference | |
| Moderate risk | 1.05 [0.44–2.53] | 0.91 | 1.85 [0.84–4.08] | 0.128 |
| Severe risk | 0.75 [0.26–2.20] | 0.603 | 1.10 [0.42–2.87] | 0.85 |
| CMMSE | | | | |

*(Continued)*

**Table 3.** (Continued)

| Variables | Sarcopenia | | Severe sarcopenia | |
|---|---|---|---|---|
| | Adjusted OR [95% C.I.] | p-value | Adjusted OR [95% C.I.] | p-value |
| Score 23 and less | 1.28 [0.54–3.04] | 0.579 | 2.26 [1.01–5.02] | 0.046 |
| Score 24 and more | Reference | | Reference | |
| Charlson Comorbidity Index | | | | |
| 0 | Reference | | Reference | |
| 1 and 2 | 0.81 [0.34–1.96] | 0.642 | 0.85 [0.38–1.89] | 0.686 |
| 3 and more | 0.95 [0.30–3.00] | 0.928 | 0.85 [0.32–2.27] | 0.745 |
| Ever infected by COVID-19 before recruitment into study | | | | |
| No | Reference | | Reference | |
| Yes | 0.89 [0.41–1.94] | 0.778 | 1.33 [0.66–2.67] | 0.423 |
| RDG group[d] | | | | |
| Others [IV antibiotics, wound care, malignancy] | 4.84 [1.07–22.03] | 0.041 | | |
| Stroke | 4.39 [0.77–23.83] | 0.095 | 1.60 [0.32–8.08] | 0.572 |
| Hip fracture | Reference | | Reference | |
| Musculoskeletal conditions | 2.57 [0.71–9.28] | 0.149 | 1.34 [0.44–4.06] | 0.602 |
| Deconditioning | 1.04 [0.20–5.42] | 0.959 | 1.74 [0.49–6.11] | 0.389 |
| MBI score on admission | | | | |
| MBI 0–60 | 1.47 [0.67–3.22] | 0.342 | 1.99 [0.96–4.14] | 0.065 |
| MBI 61–100 | Reference | | Reference | |

[a]cut-offs for waist circumference as per HPB Asia Pacific Consensus and International Diabetes Federation.

[b]cut-offs for WHC as per WHO classification of abdominal fatness.

[c]as assessed using WHO GPAQ.

[d]Rehab Diagnostic Groups.

*missing data n = 2.

reversing sarcopenia [73]. Incorporating the right mix for the best effect is currently a knowledge gap worth studying in the future.

Higher hip circumference being a surrogate measure of gluteal musculature is understandably inversely associated with sarcopenia [19,74–76]. In addition, our study finds that a higher BMI is also protective of sarcopenia similar to other studies [18,77–79].

There may be a few explanations for the above. Firstly, a higher BMI may translate to increased load-bearing requirements during daily activities increasing muscle mass [80]. Secondly, a higher BMI may be due to better nutritional intake with a study showing associations between BMI and total protein intake [81]. Thirdly, BMI is likely an inadequate surrogate measure of adiposity in older adults [82,83]. In fact, sarcopenic obesity [SO] prevalence defined with BMI underestimates those defined by body fat percentages in the same cohort in a study [78]. Interestingly, a higher BMI range, confers a paradoxical survival benefit in older adults [84,85]. A study comparing all-cause mortality within the same cohort between groups with BMI v.s. body fat percentage measurements showed no increased mortality risk in apparently obese category of BMI while those defined obese with body fat percentage measurements did [86].

Perhaps it is time to reconsider the utility of previously accepted BMI range pre-defined in Asians older adults. Despite the limitations of BMI as a surrogate measure of body fat composition, it is undeniably an easily obtainable measure in a busy setting with limited resources. However, there is a research gap in defining what is a normal BMI range correlating to sarcopenic outcomes in long term prospective prognosis studies.

It may also be timely to standardise the definition of sarcopenic obesity using body fat percentages norms in future Delphi consensus studies among content experts. Understandably, more studies are needed to correctly classify obesity using body fat percentage adjusted for sex, age and race. Aligning sarcopenic obesity diagnostic criteria will stimulate further research in this area and allow for confident synthesis of data outcomes across different studies. This will pave the way for interventional trials.

In our study, cognitively impaired participants have a higher odd of severe sarcopenia although not to sarcopenia while global literature found an association between cognitive impairment and sarcopenia [87]. We believe the loss of association of cognitive impairment with sarcopenia could be due to type 2 error.

Our study did not find an association between categories of nutritional risks and sarcopenia on multivariate analysis although there was a statistically significant association on univariate analysis. Again, this could be due to type 2 error. An alternative explanation is because of the inherent differential property of a screening tool and a full diagnostic assessment of nutritional state. For example, a study showed GLIM assessment of nutrition predicted sarcopenia among malnourished participants while a screening tool failed to do so [88]. Nonetheless various studies showed the positive effects various nutritional interventions have on sarcopenia [89,90]. Of great interest, precision nutritional prescription is a hypothesis to be tested in the future for optimization of nutritional interventions based on a combination of factors such as phenotype, dietary habits and behaviours of individuals [91].

To our knowledge, this is the first inpatient study in Singapore to report sarcopenia prevalence using AWGS diagnostic criteria. Our study affirms some of the common associations associated with sarcopenia in line with global literature. This foundational evidence will provide the possibility of developing technology or artificial intelligence driven case finding for sarcopenia in an inpatient setting via a risk calculator. This is especially important as clinicians face challenges with equipment and manpower investiture with current diagnostic criteria of sarcopenia. Our study will also provide the basis for future intervention targeted at risk factors in our population to modify sarcopenic outcomes. Based on our findings, we can start on the modifiable risk factors found in within our cohort. Overcoming barriers of physical inactivity with muti-pronged strategies such as easy access to motivational interviewing, personalized physical activity coaching and setting physical targets tailored to the individuals may promote adoption of a less sedentary lifestyle in older adults [92]. Such multi-component and complex strategies should be co-designed and structured to participant groups sociocultural context [93]. For sarcopenic participants with cognitive impairments, current studies exploring the biochemical process involved in muscle-brain cross talk are underway. Future research may crystallize treatments targeting these processes especially in those at highest risk of cognitive impairments [94].

Some other strengths of our study include a reasonably large number of recruitments with low dropouts and missing data, as well as the consecutive sampling methodology for all patients admitted to OCH to achieve a more representative sample of the population.

## Limitations

This study has several limitations. Our single site study participants may not be reflective of inpatients in other post-acute settings in other countries. Our findings cannot be generalized to those with participants characteristics that were excluded as per study protocol. Thirdly, we do not have data on the length of stay each participant spent in the acute hospital prior to their admission into our community hospital although we will generally screen all patients to ensure acute conditions have resolved prior to admissions. The uncertainty in the length of stay in

acute hospital prior to admission may be a potential bias in our study. Additionally, causal relationship of the associated factors cannot be established as it is a cross-sectional study.

Our study estimated appendicular lean mass using the InBody S10 BIA which was not validated locally. However, the use of a multifrequency BIA is an accepted method of measurement by AWGS2019 with gender-specific cut-offs. The team utilised the InBody S10 for its ability to perform measurements in a supine position, enabling assessment among participants with functional limitations. Additionally, utilisation of a BIA machine will allow for comparative data in future interventional studies spanning across inpatient to the community, where access to DXA may be difficult.

The Jamar hydraulic hand dynamometer is the recommended brand for assessing handgrip strength, however this study utilised the Baseline hydraulic hand dynamometer instead for logistical reasons as multiple sets were readily available and routinely used by the rehabilitation department therapists, who were the assessors of handgrip for this study. The Baseline dynamometer has been validated against the Jamar dynamometer to measure equivalently and hence can be used interchangeably [95].

## Conclusions

Our study found a prevalence of 54% of sarcopenia among inpatient older adults. We also found that participants with increased age, male gender, and a lower physical activity level were more likely to be sarcopenic, while those who had a higher hip circumference and higher BMI of 27.5 and above were less likely to be sarcopenic. In addition, cognitively impaired participants were more likely to be severely sarcopenic. There is an urgent need to address the burgeoning burden with sarcopenia and identify patients at higher risk of sarcopenia in post-acute settings in Singapore for early intervention.

## Author Contributions

**Conceptualization:** Lian Leng Low, Junjie Aw.

**Data curation:** Charmaine Tan You Mei, Doris Lim Yanshan, Siew Van Koh, Ganeshan Karthikeyan, Xuan Lin Low, Hui Yi Quek, Andrea Ong Shuyi.

**Formal analysis:** Charmaine Tan You Mei, Sharna Seah Si Ying, Olivia Xia Jiawen, Junjie Aw.

**Funding acquisition:** Charmaine Tan You Mei, Sharna Seah Si Ying.

**Methodology:** Charmaine Tan You Mei, Sharna Seah Si Ying, Olivia Xia Jiawen, Lian Leng Low, Junjie Aw.

**Project administration:** Charmaine Tan You Mei, Doris Lim Yanshan, Siew Van Koh, Ganeshan Karthikeyan, Xuan Lin Low, Hui Yi Quek, Andrea Ong Shuyi.

**Resources:** Charmaine Tan You Mei, Sharna Seah Si Ying, Doris Lim Yanshan, Siew Van Koh, Ganeshan Karthikeyan, Olivia Xia Jiawen, Xuan Lin Low, Hui Yi Quek, Andrea Ong Shuyi.

**Software:** Charmaine Tan You Mei, Sharna Seah Si Ying, Olivia Xia Jiawen, Junjie Aw.

**Supervision:** Siew Van Koh, Ganeshan Karthikeyan, Andrea Ong Shuyi, Junjie Aw.

**Validation:** Charmaine Tan You Mei, Olivia Xia Jiawen, Xuan Lin Low, Hui Yi Quek, Andrea Ong Shuyi, Lian Leng Low.

**Visualization:** Lian Leng Low, Junjie Aw.

**Writing – original draft:** Charmaine Tan You Mei, Junjie Aw.

**Writing – review & editing:** Charmaine Tan You Mei, Sharna Seah Si Ying, Olivia Xia Jiawen, Lian Leng Low, Junjie Aw.

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
