## [Decision Letter · Decision Letter 0]

3 Oct 2023

PONE-D-23-27709Prevalence and factors associated with sarcopenia among older adults in a post-acute hospital in SingaporePLOS ONE

Dear Dr. Aw,

Thank you for submitting your manuscript to PLOS ONE. After careful consideration, we feel that it has merit but does not fully meet PLOS ONE’s publication criteria as it currently stands. Therefore, we invite you to submit a revised version of the manuscript that addresses the points raised during the review process.

We look forward to receiving your revised manuscript.

Kind regards,

Antimo Moretti

Academic Editor

PLOS ONE

Journal Requirements:

Reviewers' comments:

Reviewer's Responses to Questions

**Comments to the Author**

1. Is the manuscript technically sound, and do the data support the conclusions?

Reviewer #1: Yes

Reviewer #2: Yes

2. Has the statistical analysis been performed appropriately and rigorously? 

Reviewer #1: Yes

Reviewer #2: Yes

3. Have the authors made all data underlying the findings in their manuscript fully available?

Reviewer #1: Yes

Reviewer #2: Yes

4. Is the manuscript presented in an intelligible fashion and written in standard English?

Reviewer #1: Yes

Reviewer #2: Yes

5. Review Comments to the Author

Reviewer #1: Dear authors,

you performed an interesting study about the prevalence of sarcopenia in a post acute population. The manuscript is quite well written and easy to read. However, some methodological issue could be better explain and improved and discussione could be better argued. Please, address the following issues.

Line 81: please, do not start a sentence with a number. It could be more readably write in letter the number

Line 106-108: Type 2 diabetes mellitus is associated not only with sarcopenia, but even with osteosarcopenia as reported in a recent publication. Please consider the article doi: 10.52965/001c.38570

Line 163-164: please explain why you prefer to use GPAQ and not IPAQ questionnaire.

Line 215-216: please explain why you prefer to use BIA and not DXA

Line 336-339: please better explain in detail the biological mechanisms underlying the onset of sarcopenia in this population.

In discussion, please make mention about some potential pharmacological and non-pharmacological approach to prevent or treat the studied condition.

Reviewer #2: Congratulations on the good work. I read the manuscript with interest.

I would like to share some doubts with the authors and I hope they can be clarified.

1) There are key points in the text, I recommend inserting keywords

2) it might be useful to summarize the risk factors associated with sarcopenia in the table

3) In the introduction I recommend listing the diagnostic criteria of sarcopenia

4) In the materials and methods it is necessary to specify the study design and indicate whether it has been approved by the ethics committee

5) I recommend including a legend for the many abbreviations

6) I recommend reviewing the references in the materials and methods (some do not correspond e.g. 44)

7) I recommend calculating the sample size (population)

8) In question the definition "high prevalence" is overused. I recommend being more precise and indicating the prevalence

9) To improve the discussion I recommend consulting the following references:

doi: 10.1177/1759720X231152648; doi: 10.1007/s40520-021-01977-x.

6. PLOS authors have the option to publish the peer review history of their article (what does this mean?). If published, this will include your full peer review and any attached files.

Reviewer #1: No

Reviewer #2: No

---

## [Author Response · Author response to Decision Letter 0]

14 Oct 2023

Dear Reviewers,

Please kindly refer to the attached MS word document: "response to reviewers v2" for our detailed response.

Warm regards,

Joshua Junjie Aw

---

## [Decision Letter · Decision Letter 1]

12 Dec 2023

PONE-D-23-27709R1Prevalence and factors associated with sarcopenia among older adults in a post-acute hospital in SingaporePLOS ONE

Dear Dr. Aw,

Thank you for submitting your manuscript to PLOS ONE. After careful consideration, we feel that it has merit but does not fully meet PLOS ONE’s publication criteria as it currently stands. Therefore, we invite you to submit a revised version of the manuscript that addresses the points raised during the review process.

We look forward to receiving your revised manuscript.

Kind regards,

Antimo Moretti

Academic Editor

PLOS ONE

Journal Requirements:

Reviewers' comments:

Reviewer's Responses to Questions

**Comments to the Author**

1. If the authors have adequately addressed your comments raised in a previous round of review and you feel that this manuscript is now acceptable for publication, you may indicate that here to bypass the “Comments to the Author” section, enter your conflict of interest statement in the “Confidential to Editor” section, and submit your "Accept" recommendation.

Reviewer #2: All comments have been addressed

Reviewer #3: (No Response)

2. Is the manuscript technically sound, and do the data support the conclusions?

Reviewer #2: Yes

Reviewer #3: Yes

3. Has the statistical analysis been performed appropriately and rigorously? 

Reviewer #2: Yes

Reviewer #3: Yes

4. Have the authors made all data underlying the findings in their manuscript fully available?

Reviewer #2: Yes

Reviewer #3: Yes

5. Is the manuscript presented in an intelligible fashion and written in standard English?

Reviewer #2: Yes

Reviewer #3: Yes

6. Review Comments to the Author

Reviewer #2: I thank the editor for giving me the opportunity to review the manuscript. The authors followed the instructions, however some aspects remain to be clarified.

1) The introduction states a prevalence of sarcopenia between the ages of 21 and 90. In reality, one of the requirements for sarcopenia is age over 65 years. I recommend not considering the study because it is contradictory

2) Results: to affirm that there is a correlation between sarcopenia and male sex you should compare sarcopenic males and females (not non-sarcopenic males and sarcopenic males). I advise you to explain better or modify.

Reviewer #3: Dear Authors,

In this cross-sectional study, you aimed to determine the prevalence of sarcopenia, severe sarcopenia and their associations in an inpatient post-acute hospital in Singapore.

The paper is well written and investigate a relevant topic, nevertheless there are some minor issues to be addressed.

Title, abstract, introduction: clear and well targeted.

Methods:

Did you consider the period lenght that each patient spent hospitalized in an acute hospital? May this influence the sarcopenia prevalence? May be this a limit of the study?

Results, discussion, conclusion: well described.

7. PLOS authors have the option to publish the peer review history of their article (what does this mean?). If published, this will include your full peer review and any attached files.

Reviewer #2: No

Reviewer #3: No

---

## [Author Response · Author response to Decision Letter 1]

29 Dec 2023

Please refer to the MS Word document for the response to the reviewers. Also copying and appending the response here for easy reference:

Review Comments to the Author

Reviewer #2: I thank the editor for giving me the opportunity to review the manuscript. The authors followed the instructions, however some aspects remain to be clarified.

Author’s reply: We thank reviewer #2 for spending time to review our manuscript. We will attempt to reply point by point to the queries raised.

1) The introduction states a prevalence of sarcopenia between the ages of 21 and 90. In reality, one of the requirements for sarcopenia is age over 65 years. I recommend not considering the study because it is contradictory

Author’s reply: We agree that the article cited in our study above in line 105 has methodological flaws regarding its inclusion criteria for age in their study. Henceforth we have adopted the reviewer #2 viewpoint that this article should not be cited in our study. We have since removed it. The changes can be found under tracked changes in line 105.

2) Results: to affirm that there is a correlation between sarcopenia and male sex you should compare sarcopenic males and females (not non-sarcopenic males and sarcopenic males). I advise you to explain better or modify.

Author’s reply: We thank Reviewer #2 for your comment. We apologize for any misunderstanding. We have since clarified further in the manuscript line 323-328 under “tracked changes” to improve the readability and interpretation of table 1 and table 2. To affirm correlation between sarcopenia and male sex, we had compared sarcopenic males versus females. Males have a higher odd of sarcopenia compared to females with odds of 2.34 and 2.8 on univariate and multivariate analyses respectively. 

Reviewer #3: Dear Authors,

In this cross-sectional study, you aimed to determine the prevalence of sarcopenia, severe sarcopenia and their associations in an inpatient post-acute hospital in Singapore.

The paper is well written and investigate a relevant topic, nevertheless there are some minor issues to be addressed.

Author’s reply: We thank Reviewer #3 for your encouraging comment!

Title, abstract, introduction: clear and well targeted.

Methods:

Did you consider the period lenght that each patient spent hospitalized in an acute hospital? May this influence the sarcopenia prevalence? May be this a limit of the study?

Results, discussion, conclusion: well described.

Author’s reply: We thank Reviewer #3 for your comment. We acknowledge that the length of time spent hospitalized in the acute hospital may influence the sarcopenia prevalence and this is a limitation of our study. We have adopted your suggestion and append this limitation in line 472-476

---

## [Editor Report · Decision Letter 2]

10 Jan 2024

Prevalence and factors associated with sarcopenia among older adults in a post-acute hospital in Singapore

PONE-D-23-27709R2

Dear Dr. Junjie Aw,

We’re pleased to inform you that your manuscript has been judged scientifically suitable for publication and will be formally accepted for publication once it meets all outstanding technical requirements.

Kind regards,

Antimo Moretti

Academic Editor

PLOS ONE

---

## [Editor Report · Acceptance letter]

19 Jan 2024

PONE-D-23-27709R2 

PLOS ONE

Dear Dr. Aw, 

I'm pleased to inform you that your manuscript has been deemed suitable for publication in PLOS ONE. Congratulations! Your manuscript is now being handed over to our production team.

Kind regards, 

on behalf of

Prof. Dr. Antimo Moretti 

Academic Editor

PLOS ONE